# Spinal Muscular Atrophy: Diagnosis, Incidence, and Newborn Screening in Japan

**DOI:** 10.3390/ijns7030045

**Published:** 2021-07-20

**Authors:** Tomokazu Kimizu, Shinobu Ida, Kentaro Okamoto, Hiroyuki Awano, Emma Tabe Eko Niba, Yogik Onky Silvana Wijaya, Shin Okazaki, Hideki Shimomura, Tomoko Lee, Koji Tominaga, Shin Nabatame, Toshio Saito, Takashi Hamazaki, Norio Sakai, Kayoko Saito, Haruo Shintaku, Kandai Nozu, Yasuhiro Takeshima, Kazumoto Iijima, Hisahide Nishio, Masakazu Shinohara

**Affiliations:** 1Department of Pediatric Neurology, Osaka Women’s and Children’s Hospital, 840 Murodocho, Izumi 594-1101, Japan; kimizu@wch.opho.jp; 2Department of Gastroenterology and Endocrinology, Osaka Women’s and Children’s Hospital, 840 Murodocho, Izumi 594-1101, Japan; idas@wch.opho.jp; 3Department of Pediatrics, Ehime Prefectural Imabari Hospital, 4-5-5 Ishiicho, Imabari 794-0006, Japan; kentaro206@gmail.com; 4Department of Pediatrics, Kobe University Graduate School of Medicine, 7-5-1 Kusunoki-cho, Kobe 650-0017, Japan; awahiro@med.kobe-u.ac.jp (H.A.); nozu@med.kobe-u.ac.jp (K.N.); iijima@med.kobe-u.ac.jp (K.I.); 5Department of Community Medicine and Social Healthcare Science, Kobe University Graduate School of Medicine, 7-5-1 Kusunoki-cho, Kobe 650-0017, Japan; niba@med.kobe-u.ac.jp (E.T.E.N.); yogik.onky@gmail.com (Y.O.S.W.); mashino@med.kobe-u.ac.jp (M.S.); 6Department of Pediatric Neurology, Children’s Medical Center, Osaka City General Hospital, 2-13-22 Miyakojimahondori, Osaka 534-0021, Japan; sokazaki2009@gmail.com; 7Department of Pediatrics, Hyogo College of Medicine, 1-1 Mukogawacho, Nishinomiya 663-8501, Japan; shimomura.ped@gmail.com (H.S.); leeleetomo@me.com (T.L.); ytake@hyo-med.ac.jp (Y.T.); 8Department of Pediatrics, Osaka University Graduate School of Medicine, 2-2 Yamadaoka, Suita 565-0871, Japan; yasuhito@ped.med.osaka-u.ac.jp (K.T.); nabatames@ped.med.osaka-u.ac.jp (S.N.); 9Division of Child Neurology, Department of Neurology, National Hospital Organization Osaka Toneyama Medical Center, 5-1-1 Toneyama, Toyonaka 560-8552, Japan; saito.toshio.cq@mail.hosp.go.jp; 10Department of Pediatrics, Osaka City University Graduate School of Medicine, 1-4-3 Asahi-machi, Osaka 545-8585, Japan; hammer@med.osaka-cu.ac.jp (T.H.); shintakuh@med.osaka-cu.ac.jp (H.S.); 11Child Healthcare and Genetic Science Laboratory, Division of Health Sciences, Osaka University Graduate School of Medicine, 2-2 Yamadaoka, Suita 565-0871, Japan; norio@ped.med.osaka-u.ac.jp; 12Institute of Medical Genetics, Tokyo Women’s Medical University, 8-1 Kawadacho, Tokyo 162-0054, Japan; saito.kayoko@twmu.ac.jp; 13Hyogo Prefectural Kobe Children’s Hospital, 1-6-7 Minatojima Minamimachi, Kobe 650-0047, Japan; 14Faculty of Medical Rehabilitation, Kobe Gakuin University, 518 Arise Ikawadani-cho, Kobe 651-2180, Japan

**Keywords:** spinal muscular atrophy, *SMN1*, deletion, incidence, newborn screening

## Abstract

Spinal muscular atrophy (SMA) is a genetic neuromuscular disorder that causes degeneration of anterior horn cells in the human spinal cord and subsequent loss of motor neurons. The severe form of SMA is among the genetic diseases with the highest infant mortality. Although SMA has been considered incurable, newly developed drugs—nusinersen and onasemnogene abeparvovec—improve the life prognoses and motor functions of affected infants. To maximize the efficacy of these drugs, treatments should be started at the pre-symptomatic stage of SMA. Thus, newborn screening for SMA is now strongly recommended. Herein, we provide some data based on our experience of SMA diagnosis by genetic testing in Japan. A total of 515 patients suspected of having SMA or another lower motor neuron disease were tested. Among these patients, 228 were diagnosed as having SMA with survival motor neuron 1 (*SMN1*) deletion. We analyzed the distribution of clinical subtypes and ages at genetic testing in the *SMN1*-deleted patients, and estimated the SMA incidence based on data from Osaka and Hyogo prefectures, Japan. Our data showed that confirmed diagnosis by genetic testing was notably delayed, and the estimated incidence was 1 in 30,000–40,000 live births, which seemed notably lower than in other countries. These findings suggest that many diagnosis-delayed or undiagnosed cases may be present in Japan. To prevent this, newborn screening programs for SMA (SMA-NBS) need to be implemented in all Japanese prefectures. In this article, we also introduce our pilot study for SMA-NBS in Osaka Prefecture.

## 1. Introduction

Spinal muscular atrophy (SMA) is a genetic neuromuscular disorder that causes degeneration of anterior horn cells in the human spinal cord and subsequent loss of motor neurons [1]. According to a previous report, it has a prevalence of approximately 1–2 per 100,000 individuals and an incidence of around 1 in 10,000 live births [2]. Two SMA-related genes mapped to chromosome 5q13, survival motor neuron 1 (*SMN1*) and survival motor neuron 2 (*SMN2*) [3], which are highly homologous, were reported in 1995. *SMN1* is now considered as a gene causative of SMA. More than 90% of SMA patients are homozygous for *SMN1* deletion, while the rest are compound heterozygous for a deleted *SMN1* allele and a mutated *SMN1* allele [3]. In contrast, *SMN2* is considered to be a modifying factor of the SMA phenotype because a higher copy number of *SMN2* may be related to a milder SMA phenotype [4].

SMA is clinically divided into five subtypes [5]: Type 0 (the most severe form with onset in the prenatal period; severe respiratory problems after birth, and, typically, death within weeks of birth), Type I (Werdnig–Hoffmann disease; a severe form with onset before 6 months of age; the inability to sit unsupported), Type II (Dubowitz disease; an intermediate form with onset before 18 months of age; the ability to sit unaided but not to stand or walk), Type III (Kugelberg–Welander disease; a mild form with onset after 18 months of age; the ability to stand and walk unaided), and Type IV (the mildest form with onset after 30 years of age). SMA type I is a genetic disease with high infant mortality [6]. Many patients with SMA type I die of respiratory insufficiency by 2 years of age, when respiratory support is not available [7]. Meanwhile, patients with types II, III and IV are also forced to lead lives with limited motor function, showing various levels of severity.

Until recently, SMA was considered to be incurable. However, treatments for this disease are emerging. In 2016, the United States Food and Drug Administration (FDA) approved nusinersen (Spinraza^®^, Biogen, Cambridge, MA), the first drug designated to treat SMA. Nusinersen is an *SMN2*-directed antisense oligonucleotide drug, which alters *SMN2′*s pre-mRNA splicing pattern to produce more SMN protein in the motor neuron cells in SMA [8]. In 2019, the FDA approved onasemnogene abeparvovec (Zolgensma^®^, AveXis Inc, Bannockburn, IL) as the second drug for treating SMA. Onasemnogene abeparvovec is an adeno-associated viral vector-based gene therapy designed to deliver a functional copy of the human *SMN* gene to the motor neuron cells of SMA patients [9]. The Japanese Ministry of Health, Labour and Welfare approved nusinersen in 2017 and approved onasemnogene abeparvovec in 2020. These new drugs improve the life prognoses and motor functions of infants affected by SMA.

According to early reports of clinical trials, nusinersen and onasemnogene abeparvovec helped patients to reach milestones in their motor function development and increased their likelihood of survival [10,11]. The most recent report of the clinical trials with these drugs showed that early treatment, especially at the pre-symptomatic stage, resulted in better outcomes in SMA patients; even SMA type I patients became able to stand and walk [12,13].

Delayed diagnosis leads to delayed treatment, resulting in limited effects on the clinical phenotype [10,11]. Thus, newborn screening (NBS) for SMA is now strongly recommended. As more than 90% of SMA patients are homozygous for *SMN1* deletion, as mentioned above, the presence or absence of *SMN1* can be a good marker for SMA screening. In Japan, NBS programs for SMA have just started in some areas, including Osaka, Hyogo, Chiba, Aichi, and Kumamoto prefectures. However, at present, the number of infants tested in the screening programs is insufficient to obtain an estimate of the incidence of SMA in Japan.

According to a global overview of the current situation of NBS for SMA (SMA-NBS) [14], nine SMA-NBS programs performed in various countries have so far detected 288 newborns with SMA out of 3,674,277 newborns screened. The annual proportion of newborns to be screened for SMA in the coming years is expected to increase steadily [14]. Our SMA-NBS programs will cover all newborns throughout Japan in the near future.

In this article, we first report our experience of genetic diagnosis based on testing for *SMN1* deletion. Second, we present the estimated incidence of SMA based on data from Osaka and Hyogo prefectures, Japan. Third, we describe a pilot study for SMA-NBS in Osaka Prefecture and discuss the need to implement SMA-NBS programs throughout Japan.

## 2. Patients and Methods

### 2.1. Diagnosis of SMA

A total of 515 patients from 36 out of 47 prefectures in Japan were referred to the Department of Community Medicine and Social Healthcare Science, Kobe University Graduate School of Medicine, in the period from 1996 to 2019. These patients were suspected of having SMA or another lower motor neuron disease (LMND). Their ages varied from several days after birth to 63 years. Infants, toddlers, and children presented with such indications as delayed developmental milestones, respiratory problems, and muscle weakness. Meanwhile, adult patients showed symptoms including walking disability and muscle weakness.

Prior to genetic analysis, written informed consent was obtained from the patients or their parents/guardians. All procedures were reviewed and approved by the Ethics Committee of Kobe University Graduate School of Medicine, and were performed in accordance with the ethical standards laid down in the Declaration of Helsinki.

Genetic testing was performed according to PCR-enzyme digestion [15] and/or multiplex ligation-dependent probe amplification analysis (MLPA) methods [16]. Copy numbers of *SMN1* and *SMN2* were determined in accordance with the methods of Harada et al. [17] and Tran et al. [18], and the MLPA method.

### 2.2. Implementation of SMA-NBS

In Osaka Prefecture, a pilot study for SMA-NBS was initiated in February 2021. This pilot study was approved by the Institutional Review Board of Osaka Women’s and Children’s Hospital. After obtaining written informed consent from the parents, DBS samples were collected from their offspring within 4–6 days after birth at the maternity hospital. Then, they were sent to Osaka Women’s and Children’s Hospital, and SMA-NBS was performed there using 5′–3′ exonuclease-based real-time PCR with fluorescent probes [19].

### 2.3. Statistical Analysis

To compare the *SMN2* copy numbers among SMA subtypes, Welch’s *t*-test was used, and to compare the proportions of confirmed diagnosis at the proper timing between SMA type I and type II, the chi-squared test was used. For these analyses, we used Microsoft Excel with the add-in software Statcel 4 (The Publisher OMS Ltd., Tokyo, Japan). A *p*-value less than 0.05 was considered to indicate statistical significance.

The incidence of SMA was defined as the number of newborn infants with the disease in a year in the population, and was expressed as the number of affected infants per 100,000 live births. The 95% confidence intervals of the incidence were calculated based on the Poisson distribution using Microsoft Excel. Population data were provided by the Statistics Bureau, Ministry of Internal Affairs and Communications of Japan.

## 3. Results

### 3.1. Genetic Analysis of Patients Suspected of Having SMA

#### 3.1.1. *SMN1* Deletion Test

Among 515 patients who were referred to our laboratory in the period from 1996 to 2019, we confirmed 228 cases as having SMA with homozygous *SMN1* deletion [15]. The remaining 287 cases retained at least one *SMN1* copy. Subsequently, 33 out of the 287 patients with SMA- or LMND-like symptoms were shown to carry only one *SMN1* copy, while 13 out of these 33 carried a deleterious *SMN1* mutation causing SMA. Our data demonstrated 44.3% of the 515 patients (228/515) carried a homozygous *SMN1* deletion, 2.5% (13/515) carried an intragenic mutation (or a subtle mutation) and an *SMN1* deletion, whereas 53.2% (274/515) remained undiagnosed at the genetic level. Thus, *SMN1*-related SMA may account for half of the patients with SMA-like or LMND-like symptoms. To clarify the causative gene abnormalities in the patients with non-*SMN1*-related SMA, targeted resequencing analysis using next-generation sequencing technology may be essential [20,21].

#### 3.1.2. Distribution of SMA Subtype and *SMN2* Copy Number

We determined the subtype distribution in the patients tested in our laboratory. Here, we used the data of 221 SMA patients with homozygous *SMN1* deletion for whom clinical information was available. Among these 221 patients tested in our laboratory, 42.1% (93/221) were diagnosed with SMA type I, 32.1% (71/221) with type II, 20.8% (46/221) with type III, and 5.0% (11/221) with type IV. Three families with affected siblings were included in our database. Each family had two affected siblings with subtype concordance (type II or type III).

Next, we determined the *SMN2* copy number in 204 out of 221 SMA patients with homozygous *SMN1* deletion (83 out of 93 patients with SMA type I, 70 out of 71 patients with SMA type II, 40 out of 46 patients with SMA type III, and 11 out of 11 patients with SMA type IV). A high *SMN2* copy number modifies the phenotype of SMA patients with homozygous deletion of *SMN1* [5]. Our database of SMA patients with homozygous *SMN1* deletion supported the conventional observation of a low *SMN2* copy number resulting in a severe phenotype and a high copy number potentially being related to a milder one (Table 1). Patients with SMA type I usually carry only two copies of *SMN2*, while SMA type II is usually associated with three copies. SMA type III patients have three to four copies, and SMA type IV patients usually have four or more copies. A high *SMN2* copy number may improve the survival outcomes and motor function.

### 3.2. Age of Genetic Testing among SMA Patients

Since *SMN1* was identified as an SMA-causing gene in 1995, genetic testing, especially testing for the *SMN1* deletion, has been widely used to confirm the diagnosis of SMA [5]. The age at genetic testing of the patients referred to our laboratory is shown in Table 2. In this analysis, we used the data of 142 SMA patients with homozygous *SMN1* deletion (84 type I, 43 type II, and 15 type III). The remaining SMA patients were excluded because the onset age information was missing. These patients were born between January 1996 and September 2018. The mean ages at genetic testing were 11.0 months old (standard deviation (SD) ± 23.7) for type I, 77.3 months old (SD ± 79.9) for type II, and 85.1 months old (SD ± 79.1) for type III.

The exact ages of onset of many patients were not available in this study. Thus, we could not determine the exact duration between onset and genetic testing. Instead, we calculated the proportions of cases with a “proper,” “slightly delayed,” or “notably delayed” timing of genetic testing for the confirmed diagnosis of SMA type I and type II, as shown in Table 2. As for SMA type III, we excluded it from the “timing of genetic testing” analysis because it is a late-onset and slowly progressive disease.

The definitions of “proper”, “slightly delayed,” and “notably delayed” timing were as follows: (1) for SMA type I, “proper” timing of diagnosis is within 6 months after birth, “slightly delayed” timing is between 6 and 12 months, and “notably delayed” timing is more than 12 months; (2) for SMA type II, “proper” timing of diagnosis is earlier than 18 months, “slightly delayed” timing is between 18 and 24 months, and “notably delayed” timing is more than 24 months.

Overall, confirmed diagnosis of SMA was slightly delayed or notably delayed in 63 out of 127 (50.0%) patients (29 out of 84 type I patients, and 34 out of 43 type II patients). Only 20.9% of type II patients were diagnosed at the proper timing, which was a markedly lower rate than that of type I patients (65.5%). There was a significant difference in the number of confirmed diagnoses at the proper timing between type I and type II patients (*p* < 0.01).

### 3.3. Epidemiological Analysis of SMA in Osaka and Hyogo Prefectures

We estimated the incidence of SMA using the number of SMA-affected infants who were born or SMA-affected fetuses who were aborted in Osaka and Hyogo prefectures from 2007 to 2016 and diagnosed as having SMA by genetic testing within this period (Table 3).

The number of newborn infants with SMA (including types I, II, and III) was 28 out of 1,197,156 live births. We thus estimated that the incidence of SMA was 2.34 per 100,000 live births (95% CI: −0.66, 4.53)—that is, ~1 in 40,000. We also had information on aborted fetuses with a prenatal diagnosis of SMA. When this number was added to the number of newborns with SMA, the estimated incidence of SMA was 3.09 per 100,000 live births (95% CI: −0.36, 5.20)—that is, ~1 in 30,000.

The number of newborn infants with SMA type I was 14 out of 1,197,156 live births. We thus calculated that the estimated incidence of SMA type I was 1.08 per 100,000 live births (95% CI: −0.95, 3.20)—that is, ~1 in 100,000. When the number of aborted infants with a prenatal diagnosis of SMA type I was added to the number of newborn infants with SMA, the estimated incidence of SMA type I was 1.32 per 100,000 live births (95% CI: −0.84, 3.92)—that is, ~1 in 80,000. Here, the diagnosis of SMA type I in the aborted fetuses was based on the clinical subtype of the patients in the same family. We analyzed all cases of SMA-affected fetuses and their affected sibling in our laboratory.

### 3.4. Implementation of SMA-NBS in Osaka Prefecture

More than 10,000 new DBS samples from neonates born in Osaka Prefecture were tested in the pilot study for an SMA-NBS program as of 17 May 2021. The assay tested for the presence/absence of *SMN1*. All DBS samples tested negative for SMA.

## 4. Discussion

### 4.1. SMA Subtype and SMN2 Copy Number in Japanese SMA Patients

Almost all data reported from groups around the world, including ours, show similar tendencies (Table 4). Specifically, type I patients predominate (40%–60% of all SMA patients), while type IV patients are very rare (less than 5% of all SMA patients) in all populations. Roughly speaking, half of SMA patients are type I, while the other half are types II and III.

A high *SMN2* copy number modifies the phenotype of SMA patients with homozygous deletion of *SMN1* [5]. Our database of SMA patients with homozygous *SMN1* deletion supported the conventional observation that a low *SMN2* copy number results in a severe phenotype and a high copy number may be related to milder SMA phenotypes (Table 1). However, we did not observe such a tendency in all SMA patients with an intragenic *SMN1* mutation in our previous studies [22,23].

Among the SMA patients with an intragenic mutation, some with a milder phenotype carried only a single *SMN2* copy, while others with a severe phenotype carried three *SMN2* copies. For patients with an intragenic *SMN1* mutation, we cannot conclude that clinical severity is inversely correlated with the *SMN2* copy number. The locations and types of intragenic *SMN1* mutations may make more significant contributions to the clinical phenotype than the *SMN2* copy number.

### 4.2. Age at Genetic Testing among Japanese SMA Patients

Upon comparison with the findings in previous reports from other countries, we conclude that SMA diagnosis may still be delayed in Japan. According to a review of 21 studies reported in the literature by Lin et al. [32] in 2015, the weighted mean ages of genetic diagnosis were 6.3 months old (SD ± 2.2), 20.7 months old (SD ± 2.6), and 50.3 months old (SD ± 12.9), for types I, II, and III, respectively. Pera et al. also reported a study on the age of genetic diagnosis of SMA in Italy [33]. The cohort included 480 patients (191 type I, 210 type II, and 79 type III). The mean ages of genetic diagnosis were 4.70 months old (SD ± 2.82) for type I, 15.6 months old (SD ± 5.88) for type II, and 4.34 years old (SD ± 4.01) for type III.

In our study in Japan, a confirmed diagnosis of SMA type II was often delayed. Only 20.9% of type II patients were diagnosed at the proper timing, while 65.5% of type I patients were diagnosed at the proper timing. This may reflect the respiratory-stable condition of infants or toddlers with type II SMA, which does not require an urgent diagnosis. Specifically, SMA type II infants are usually respiratory-stable and do not need respiratory support. In contrast, infants with type I SMA often suffer from respiratory insufficiency.

### 4.3. Incidence of SMA in Hyogo and Osaka Prefectures

Few studies on the incidence of SMA in Japan have been performed. According to a survey by Imaizumi based on death certificate records [34], the incidence of SMA type I between 1979 and 1996 was estimated to be 1.2 per 100,000 live births in Japan. This report used only clinically diagnosed cases, and some cases with SMA type I were not included because other disease names were likely used on the death certificate.

According to a report based on a survey by Okamoto et al. [29], the incidence of infantile SMA (type I) was estimated to be 2.70 per 100,000 live births (95% CI: 0.1–5.4) on Shikoku Island, Japan, between 2011 and 2015.

We also estimated the incidence of SMA using the number of SMA-affected infants who were born or SMA-affected fetuses who were aborted in Hyogo and Osaka prefectures from 2007 to 2016 (Table 3). Based on our data in this study, the incidences of SMA (total) and SMA type I were estimated to be 3.09 per 100,000 live births (95% CI: −0.36, 5.20) and 1.32 per 100,000 live births (95% CI: −0.84, 3.92), respectively.

The incidence of SMA among Asian populations is lower than that in Western countries. Belter et al., who analyzed the Cure SMA membership database, described in their report [28] that, “Hispanics and Asians have a lower projected SMA incidence than the general population, and it follows that states with a higher proportion of Hispanics and Asians would have a lower overall incidence of SMA than the general population.”

Even so, our estimate of the incidence of SMA in Japan was notably lower than the data reported from another Asian country, Taiwan [39] (Table 5). We are thus concerned that many SMA patients may be overlooked or misdiagnosed in Japan.

### 4.4. Initiation of NBS for SMA in Japan

In Japan, a mandatory NBS program for inherited disorders is conducted using DBS samples from infants. All municipalities have mandatory with defined opt-out policies for parents. At the initiation of nationwide implementation of this program in 1977, only five diseases (phenylketonuria, galactosemia, maple syrup urine, homocystinuria, and histidinemia) were screened.

Since tandem mass spectrometry analysis was introduced into the NBS program as a first-line screening methodology in the early 2000s, many inborn errors of metabolism have been added to the list of target diseases for primary screening. Twenty diseases are now included among the screening targets.

Recently, many families of patients suffering from inherited disorders such as SMA, severe combined immunodeficiency disease (SCID) and lysosomal storage disease have demanded for the inclusion of such diseases in the NBS program. At the time of writing this, such expanded screening programs are about to start as optional programs in prefectures including Osaka, Hyogo, Chiba, Aichi, and Kumamoto.

We have also been engaged in developing new SMA screening technologies using DBS since 2010 [41,45,46,47]. The most sophisticated technology that we have developed to date is an allele-specific real-time PCR with short primers (10–12 mers), which we named “modified competitive oligonucleotide priming (mCOP)-PCR” [41,47]. We applied this mCOP-PCR technology to a prospective SMA screening study using DBS samples from 4157 Japanese newborns [41]. All DBS samples tested were negative, but there were no screening failures or false positives [41]. These results indicated that our system is applicable to SMA-NBS programs in any region or country. Nonetheless, we still think that our system should be investigated and reported on further because we are aiming for implementation in all prefectures in Japan and other countries.

Note: during the submission of this manuscript, it was reported that an infant with SMA had been detected by a pilot study for the NBS program in Kumamoto Prefecture [48]. The infant was reported to be treated with onasemnogene abeparvovec.

## 5. Conclusions

In this paper, we have described our experience of SMA diagnosis and the estimated the incidence of SMA based on data from Osaka and Hyogo prefectures, Japan. Upon comparing our data to findings previously reported from other countries, the diagnosis of SMA as confirmed by genetic testing was delayed in many Japanese patients. In addition, the estimated incidence of SMA was notably lower than that reported in other countries. These results suggested that many SMA patients in Japan may be overlooked or misdiagnosed. SMA-NBS would allow SMA patients to undergo early treatment with the potential for the maximal therapeutic benefit. Thus, there is an urgent need to implement a diagnostic system incorporating SMA-NBS in Japan. In addition, if an SMA screening system such as SMA-NBS becomes available to current Japanese patients with SMA-like or LMND-like symptoms, a large proportion of such patients may be diagnosed as having SMA and could access the new therapies. Finally, we hope that SMA-NBS programs will soon be implemented in all prefectures in Japan.

## Figures and Tables

**Table 1 IJNS-07-00045-t001:** SMA types and copy number of *SMN2* in *SMN1*-deleted patients.

Copy Number	1	2	3	4	Mean ± SD
Type I (n = 83)	1	66	16	0	2.18 ± 0.64
Type II (n = 70)	0	3	67	0	2.96 ± 0.09
Type III (n = 40)	0	2	25	13	3.28 ± 0.15
Type IV (n = 11)	0	0	1	10	3.91 ± 0.58
Total (n = 204)	1	71	109	23	

Welch’s *t*-test was used to determine differences between groups. All of the differences between groups were significant with *p* < 0.01.

**Table 2 IJNS-07-00045-t002:** Age and timing of confirmed diagnosis by genetic testing.

(A) Age at Genetic Testing (Months)			
	**Mean Age (SD)**	**Median (Range)**	**Interquartile Range**
Type I (n = 84)	11.0 (23.7)	5 (0 to 182)	7
Type II (n = 43)	77.3 (79.9)	29 (13 to 262)	122
Type III (n = 15)	85.1 (79.1)	45 (22 to 239)	79
**(B) Timing at Genetic Diagnosis (Months)**			
	**Proper Timing**	**Slightly Delayed** **Timing**	**Notably Delayed** **Timing**
Type I (n = 84)	<6 m	6 to 12 m	>12 m
	55 (65.5%)	15 (17.9%)	14 (16.7%)
Type II (n = 43)	<18 m	18 to 30 m	>30 m
	9 (20.9%)	13 (30.2%)	21 (48.8%)

**Table 3 IJNS-07-00045-t003:** Incidence of SMA in Osaka and Hyogo Prefectures.

Live Birth (n = 1,197,156)	Affected Individuals		
	Infants	Fetuses	Total
No. of types I, II and III	28	9	37
Incidence *	2.34 (95%CI: −0.66, 4.53)		3.09 (95%CI: −0.36, 5.20)
No. of type I	14	7	21
Incidence *	1.08 (95%CI: −0.95, 3.20)		1.32 (95%CI: −0.84, 3.92)

* Incidence in 100,000 of population (study period, 2007 to 2016).

**Table 4 IJNS-07-00045-t004:** Subtype distribution in countries.

Country	Total Patient Number	Type I	Type II	Type III	Type IV	Unknown
Germany (1999) [24]	525 (a)	270 (51.4%)	124 (23.6%)	131 (25.0%)	*	*
Saudi Arabia (2003) [25]	121 (a)	60 (49.6%)	26 (21.5%)	35 (28.9%)	*	*
South Africa (2007) [26]	24 (a) (White)	15 (62.5%)	4 (type II & III) (16.6%)		*	5 (20.9%)
	92 (a) (Black)	48 (52.2%)	39 (types II & III) (42.4%)		*	5 (5.4%)
Malaysia (2007) [27]	24 (a)	10 (41.7%)	11 (45.8%)	3 (12.5%)	*	*
Vietnam (2008) [18]	34 (a)	13 (38.2%)	11 (32.4%)	10 (29.4%)	*	*
Spain (2018) [4]	625 (a)	272 (43.5%)	186 (29.7%)	167 (26.7%)	*	*
Cure SMA (2018) [28]	1966 (b) (Worldwide)	1021 (51.9%)	635 (32.3%)	310 (15.8%)	*	*
Japan (2019) [29]	486 (a)	164 (33.7%)	210 (43.2%)	99 (20.4%)	7 (1.4%)	6 (1.0%)
China (2020) [30]	419 (a)	177 (45.6%)	126 (27.4%)	100 (23.2%)	16 (3.8%)	*
Greece (2020) [31]	361 (a)	156 (43.2%)	93 (25.8%)	107 (29.6%)	5 (1.4%)	*
Japan (This study)	221 (a)	93 (42.1%)	71 (32.1%)	46 (20.8%)	11 (5.0%)	*

(a) Patients confirmed diagnosis by genetic testing. (b) Self-identified patients registered in the Cure SMA database, one of the largest patient-reported data repositories on SMA patients worldwide. About 59.0% of affected individuals in the U.S.A. are registered in the Cure SMA database. * There is no description in the original article.

**Table 5 IJNS-07-00045-t005:** Incidence of SMA in various countries.

(A) Incidences of SMA Based on Survey Research					
SMA Types I, II & III					
**Country**	**Study** **Period**	**Cases** **Detected**	**Live Births**	**Incidence** (**In 100,000**)	**Reference**
Sweden	1980–2006	45	531,746	8.5 (a)	(2009) [35]
Poland	1998–2005	304	2,963,783	10.3	(2010) [36]
Europe	2011–2015	3776	22,325,221	11.9 (b)	(2017) [2]
Japan (c)	2007–2016	37(d)	1,197,178	3.1	This study
SMA Type I					
**Country**	**Study** **Period**	**Cases** **Detected**	**Live Births**	**Incidence** (**In 100,000**)	**Reference**
Sweden	1980–2006	19	531,746	3.6	(2009) [35]
Estonia	1994–2003	9	129,832	6.9	(2006) [37]
Poland	1998–2005	209	2,963,783	7.1	(2010) [36]
Japan (e)	2011–2015	4	147,950	2.7	(2019) [29]
Japan (c)	2007–2016	21(d)	1,197,178	1.3	This study
**(B) Incidences of SMA Based on Newborn Screening Programs**					
SMA Types I, II & III					
**Country**	**Study Period**	**Cases**	**Live Births**	**Incidence** (**In 100,000**)	**Reference**
U.S. (Ohio)	–2009	4	40,103	10.0	(2010) [38]
Taiwan	2014–2016	7	120,267	5.8	(2017) [39]
U.S. (New York City)	2016–2017	1	3826	26.1 (f)	(2018) [40]
Japan	2018–2019	0	4157	0.0 (f)	(2019) [41]
Germany	2018–2020	43	297,163	14.5	(2021) [42]
Australia	2018–2020	18	202,388	8.9	(2021) [43]
U.S. (North Carolina)	2018–2020	1	12,065	8.3	(2021) [44]

(a) All patients were younger than 16 years old. (b) Median incidence of several countries in Europe. (c) Hyogo and Osaka Prefectures in Japan. (d) Case number included affected infants and fetuses. (e) Ehime, Kagawa, Tokushima and Kochi Prefectures in Japan (f) The number of newborns tested in the SMA-NBS program was too small to obtain precise incidence of SMA patients.

## Data Availability

The data presented in this paper are available on request from the corresponding author. Publicly available datasets were also analyzed in this study.

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
