# Peer review of "Spinal Muscular Atrophy: Diagnosis, Incidence, and Newborn Screening in Japan"

_2409-515X, 2021, doi:10.3390/ijns7030045_

Round 1

Reviewer 1 Report

This manuscript has major weaknesses and should be rejected in its current form. The manuscript was very disorganized and unclear, with large gaps in the methods, and a confusing set of conclusions that were spread across the entire document. Figures and charts were not clear, and it is sometimes difficult to tell which data are new, where they came from, and which data are from others. 

The manuscript was submitted as a review, but the authors presented new data and drew conclusions about that data. In addition, there was minimal information on the methods (and no Methods Section because it was structured as a review). For example, in section 2.7 on the prevalence of SMA, the authors state that they calculated the prevalence of SMA based on Japanese government data, but information on the methods and limitation of the data was exceedingly sparse. In some sections it was difficult to understand the source of the data. Overall, the content that should have been the Methods and Results sections were disorganized, difficult to follow, and not comprehensive. Data on the experience of SMA testing in Japan would be important to report on, but the structure and format of this manuscript made that reporting impossible to parse. A separate manuscript in a research article format, reporting on a narrower scope of data, might be meaningful but this current manuscript is untenable.

The manuscript also does not have correct or readable English. The language used was repeatedly grammatically incorrect. Scientific terms in English were often not used properly. The errors in the English were extensive. Reading the manuscript was difficult because of the language used.

The authors also include irrelevant details or use language that implies an emotional reaction or judgement. For example, in line 128 the authors state that they, “…were very much excited by their article and celebrated the discovery of an SMA causing gene.” This may be the case, but it doesn’t have a place in the manuscript. In another example in line 77-78, the authors say, “Meanwhile, patients with types 2, 3, and 4 are also forced to lead lives with limited motor function, whether it is serious or not.”

Reviewer 2 Report

The paper "Spinal muscular atrophy: diagnosis, epidemiology and newborn screening in Osaka and Hyogo prefectures, Japan" contains some interesting data however the paper could benefit from a rewrite. 

There needs to be  correction of the english especially with relation to tense current versus past tense and requirement for the word "the" or "a" for appropriate usage.

Unfortunately the writing style is not concise presenting the same concept multiple times and much of the information could be rewritten to be more succinct. For example: Sections 2.3 and 2.4 could be joined with subtype and SMN2 copy numbers in the one section.

There is also information provided that does not add value to the article. Do we really need to know about the 1995 earthquake in this article? Or do we need the detail that KSDC was not an official organisation?

Table 1 is intended to indicate the subtype distribution in countries and should be included in the introduction with the exception of the results from this study. As well a number of countries for whom references are already included are not listed in that table eg Taiwan, Australia, or USA. Possibly only quote CURE as an international database rather than individual countries

Is there overlap between the cases of SMA in reference 24 with those presented in this submission? If so, how much?

Other minor corrections depending on the rewrite

line 213: indicates patients with type 1 usually carry one or two copies of SMN1 however there is only one patient with one copy and 69 with 2 copies. This should be amended to "usually carry 2 copies"

line 215: SMA type III should be SMA type IV

There is no doubt that screening for SMA is beneficial. The long term benefits are treatment are yet to be assessed. The screening pathways including the analytical protocol will also be modified with further data.

line 381-392: The authors mention a newly developed assay which has been used for DBS from 4157 infants. This assay should be further investigated and details reported.

Round 2

Reviewer 1 Report

This manuscript is a dramatic improvement over the original submission. The English language grammatical errors were almost completely corrected, and readability is markedly improved. The manuscript is also now more correctly and clearly organized as an Article type with appropriate sections.

Some improvements to the manuscript could be made, but I support acceptance of the manuscript after revisions. Comments:

  • Line 79 mentions “types 2, 3, and 4” but the rest of the manuscript lists the types with Roman Numerals (I, II, III). Terminology needs to be consistent throughout.
  • Line 70 mentions SMA type 0. The description says “(onset in the prenatal period; severe respiratory problems after birth)”. I would add something about death early in life in order to differentiate it with line 76 where the authors state that SMA type I has high mortality.
  • Line 140: A minor point, but typically I would say “within 4-6 days after birth” rather than “within days 4-6 after birth”
  • Line 142-3: Reference for assay used at Osaka hospital?
  • Line 159: Over what dates/time period were the 515 patients referred to your laboratory
  • Line 163: The word “Finally” seems extraneous here
  • Paragraphs between lines 180-191: I would just report the #’s here, not a comparison with what is “typically” found. These two paragraphs could be combined and shortened. Indeed, much of the comparison in paragraph two about what is typically found might be better put in the discussion.
  • Line 198: I would say “…especially testing for the SMN1 deletion” rather than “…especially the SMN1 deletion test.”
  • Line 200: Why did you use the sample size of 142? That wasn’t clear to this reviewer.
  • Line 220: I would say “slightly delayed or notably delayed” rather than just “delayed.” So it would be: “Overall, confirmed diagnosis of SMA was slightly delayed or notably delayed in 63 out of 127…”
  • Section 3.3 Where did this data, especially the data on SMA-affected fetuses come from”? There needs to be info on the methods for this analysis in the methods section. Line 234 just says “We had information on aborted fetuses with a prenatal diagnosis of SMA. Where from??
  • Line 232: I would say, “The numbers of newborn infants with SMA (including types I, II, and II) were 28 out of 1,197156 live births.” Make the same correction in line 238.
  • Line 243: Do all of the aborted fetuses have a family member with a clinical subtype?
  • Line 250:” “…Osaka Prefecture were tested in the pilot study…”
  • Line: 252: I would end the sentence with the word “negative.” No need to mention false or true positives. Might also want to instead say that “All DBS samples tested negative for SMA”
  • Section 3.4: Maybe mention the assay used? Or at least that the assay tested for the presence/absence of SMN1?
  • Line 268: The phrase “inverse correlation theory” shouldn’t be in quotes and in itself is confusing. What about instead, “For patients with an intragenic SMN1 mutation, we cannot conclude that clinical severity is inversely correlated with SMN2 copy number”
  • Table: 4: The last row should say “Japan” rather than “Japanese.”
  • Line 281: The verb “must accept” seems odd here. What about instead,” …we conclude that SMA diagnosis may still be delayed in Japan.”
  • Line 284: The sentence ‘In this literature review, 21 studies were included” is duplicative.
  • Line 291-294: “This may reflect the stable condition of infants or toddlers with type II SMA…” doesn’t make sense. The infants present with clinical symptoms later. I would focus on the later onset of symptoms rather than using the “stable condition” terminology.
  • Table 5B: Include info on the North Carolina pilot published recently? Kucera, K. S., Taylor, J. L., Robles, V. R., Clinard, K., Migliore, B., Boyea, B. L., ... & Gehtland, L. M. (2021). A Voluntary Statewide Newborn Screening Pilot for Spinal Muscular Atrophy: Results from Early Check. International journal of neonatal screening7(1), 20.
  • Line 309: It should be just “Few studies…” rather than “A few studies…”
  • Line 313: Maybe “likely used”? How do you know other disease names really were used?
  • Line 314: The word “questionnaire” is duplicative. Just “survey” is sufficient.
  • Line 316: The sentence about how the data were reliable because the rate of response to the questionnaire was 100% is inaccurate. How do you know that the respondents were accurate? The sentence that begins “The data were reliable…” could be deleted. Similarly, the following sentence about “…obtained detailed clinical information, which was essential…” could be deleted.
  • Paragraph beginning at line 321: A more direct comparison of your data to other Japanese data would benefit the publication.
  • Line 323: “Based on our data…” is in a too-large font size.
  • Line 338: The sentence that starts with “More precisely, in all municipalities…” is duplicative with the previous sentence
  • Line 343: Do not put quotes around “inborn error of metabolism”. Then simply report how many total diseases are being screened for.
  • The sentence “Recently, increasing demands have been made…” is confusing and needs to be re-worded.
  • It’s not quite clear to me that the new assay you’ve developed wasn’t used for the NBS pilot.
  • Line360: I would not say “society-wide”. Maybe instead “implementation in all prefectures in Japan” or “for all of Japan”?
  • It’s unclear where the sentence in line 356 comes from. “All DBS samples tested were negative…” Was this in reference #29?
  • Line 377: The sentence “SMA screening system…” doesn’t make sense. Is it even necessary?

Reviewer 2 Report

The authors present a well written account of SMA in Japan. The authors should be congratulated for the revisions made since original submission which have significantly improved the readability of the article

One very minor suggested modification is in 3.1.2:

The authors present subtype data for 221 patients with homozygous SMN1 deletion based on clinical information available but determined SMN2 copy number for 204 with no explanation of why this was not performed for the other 17.
